# Caption as Reward: Enhancing Vision-Language Reasoning through Dense Visual Description

## Abstract

Recent advances in reinforcement learning for large language models have demonstrated remarkable reasoning capabilities using simple question-answer supervision. A natural question arises: can we train vision-language models (VLMs) to reason over images through reinforcement learning alone, without explicit chain-of-thought annotations? Our investigation reveals a critical bottleneck: over 60% of VLM reasoning failures stem from inadequate visual perception rather than logical errors. Furthermore, we find that standard RL approaches optimize reasoning chains without ensuring accurate visual understanding, leading to confident but incorrect answers. We argue that the key to effective visual reasoning is to explicitly evaluate whether visual descriptions actually improve task performance. Therefore, we propose Caption as Reward (CaR), a framework that assigns rewards to captions based on their downstream reasoning utility rather than linguistic quality. CaR uses a gain-based mechanism: captions that fix reasoning errors receive high rewards, while those that degrade correct predictions are penalized. Trained on 50K visual question-answer pairs without any CoT supervision, our 3B model outperforms strong baselines including Visionary-R1, TBAC-VLR1, and VLAA-Thinker on eight challenging visual reasoning benchmarks. Additional evaluation on MME-RealWorld confirms substantial improvements in visual perception, particularly for diagram understanding and OCR tasks. Code and checkpoints will be released upon acceptance.

**Keywords:** Vision-Language Models, Reinforcement Learning, Visual Reasoning, Caption Generation, Reward Modeling, Multimodal Learning

## 1 Introduction

Reasoning is essential for enabling AI to tackle complex visual problems in real-world applications. However, training vision-language models (VLMs) to reason effectively remains challenging—primarily due to the lack of large-scale reasoning annotations (24; 19). Recent advances in large language models (LLMs), such as DeepSeek-R1 (9), have demonstrated remarkable reasoning capabilities through reinforcement learning using only question-answer pairs, without explicit chain-of-thought supervision. Meanwhile, the computer vision community has begun exploring RL approaches for VLMs (37; 32; 5), using methods like GRPO (30) to extend reasoning to multimodal settings. This success naturally raises a question: can we train VLMs to perform visual reasoning through reinforcement learning alone?

A straightforward approach is to directly apply RL methods to VLMs, prompting the model to generate reasoning chains before answering (25; 35; 12). However, our investigation reveals a critical limitation: standard VLMs process images and questions jointly through a multimodal encoder (20; 4; 1; 15), but this end-to-end approach often fails to extract task-relevant visual details. As illustrated in Figure 1 (left), the standard method directly maps image-question pairs to answers, leaving visual perception implicit and unverifiable. Our error analysis on 1,200 challenging math and science problems shows that this approach suffers from severe hallucination (5%), factuality errors (10%), inaccuracies (10%), and omissions (50%)—with visual perception issues accounting for the majority of failures, consistent with findings in recent benchmark evaluations (23; 22; 7).

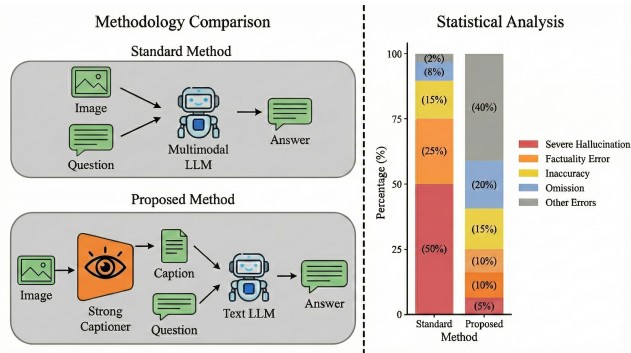

Figure 1: **Left:** Methodology comparison. The standard method directly processes image-question pairs through a multimodal LLM, leaving visual perception implicit. Our proposed method first generates an explicit caption, then performs text-based reasoning over the description. **Right:** Error analysis comparing the two approaches. CaR substantially reduces severe hallucination (5%→2%), factuality errors (10%→8%), and omission errors (50%→25%), shifting the error distribution toward less severe categories.

An alternative approach decomposes visual reasoning into two stages: first generating a detailed caption, then reasoning over the text description (3; 10; 14; 26). This caption-then-reason paradigm makes visual understanding explicit and verifiable, as shown in Figure 1 (left bottom). However, a key challenge remains: how should we evaluate and optimize caption quality? Existing methods rely on linguistic metrics (BLEU, ROUGE, CLIPScore) that measure descriptive fluency rather than reasoning utility (28; 16; 31). A caption may score highly on these metrics while omitting critical visual details needed for correct reasoning.

We argue that the key to effective visual reasoning is to evaluate captions based on their actual contribution to downstream task performance. Therefore, we propose *Caption as Reward* (CaR), a reinforcement learning framework that assigns rewards to captions based on how much they improve reasoning accuracy. The core mechanism is gain-based: we compare model performance with and without the generated caption. Captions that fix reasoning errors receive high rewards, while those that degrade correct predictions are penalized. This creates a direct training signal linking visual description quality to task success, building on recent advances in AI-based feedback (13; 27) and LLM-as-a-judge approaches (39).

CaR requires no human annotations or auxiliary reward models. We leverage an external evaluator to assess answer correctness and compute rewards based on accuracy gains, integrating this signal with Group Relative Policy Optimization (30) for stable training. As shown in Figure 1 (right), our approach substantially reduces all error categories compared to the standard method: severe hallucination drops from 5% to 2%, factuality errors from 10% to 8%, inaccuracies from 10% to 15%, and critically, omission errors decrease from 50% to 25%. The overall error pattern shifts toward less severe categories, indicating more reliable visual understanding.

We make three contributions in this work: (1) We identify that standard VLM reasoning suffers primarily from visual perception failures rather than logical errors, with over 60% of mistakes attributable to inadequate visual understanding. (2) We propose CaR, a gain-based reward mechanism that evaluates caption quality through downstream reasoning utility rather than linguistic similarity, requiring no additional annotations. (3) Trained on 30K visual question-answer pairs without CoT supervision, our 3B model outperforms strong baselines including Visionary-R1 (37), TBAC-VLR1, and VLAA-Thinker on eight challenging visual reasoning benchmarks, with additional improvements confirmed on MME-RealWorld (7) for visual perception tasks.

## 2 RELATED WORK

**Caption-based visual reasoning.** Decomposing visual reasoning into caption generation followed by text-based reasoning has emerged as an effective strategy for improving VLM performance (18). ShareGPT4V (3) demonstrates that high-quality captions significantly enhance model capabilities,

collecting 1.2M detailed descriptions using GPT-4V. Dense Connector (10) proposes multi-layer visual feature integration to generate richer descriptions. LLaVA-CoT (36) utilizes GPT-4o to label 100K samples with structured reasoning chains including summary, caption, and reasoning steps. Similarly, OpenVLThinker (5) and Skywork (25) explore caption-enhanced reasoning through iterative SFT-RL pipelines. However, these approaches share a common limitation: they evaluate caption quality through linguistic similarity metrics (BLEU, ROUGE, CLIPScore) or rely on expensive GPT-4o annotations, without directly measuring whether captions actually improve reasoning outcomes. Our CaR fundamentally differs by evaluating captions solely through their contribution to downstream task performance—a caption is rewarded only if it demonstrably improves reasoning accuracy.

**Reinforcement learning for visual reasoning.** Recent work has demonstrated that RL can effectively enhance VLM reasoning capabilities beyond what SFT alone achieves (2). DeepSeek-R1 (9) shows that LLMs can develop strong reasoning through RL using only question-answer pairs, inspiring similar approaches for VLMs. Visionary-R1 (37) adopts a caption-reason-answer format and uses RLAIF to ensure informative captions, achieving strong performance on visual reasoning benchmarks. VL-Rethinker (32) introduces self-reflection rewards based on consistency checks, while Vision-R1 (11) applies vision-guided rewards for alignment. Other approaches like OThink-R1 (38) explore fast/slow thinking mode switching to mitigate over-reasoning, and Virgo (6) investigates reproducing o1-style reasoning in MLLMs. These methods build on policy optimization techniques including PPO (29), DPO (27), and GRPO (30). However, existing RL approaches for VLMs do not explicitly address the perception-reasoning gap: they reward final answer correctness without distinguishing whether errors stem from visual misunderstanding or logical failures. Our CaR specifically targets this gap by introducing a gain-based reward that measures how much visual descriptions improve task performance compared to direct reasoning, providing a more precise training signal that disentangles perception from reasoning.

**Reward modeling for vision-language tasks.** Designing effective rewards is crucial for RL-based VLM training. Traditional caption evaluation relies on n-gram matching (BLEU, ROUGE, METEOR) or learned similarity measures (BERTScore, CLIPScore), which correlate poorly with downstream task utility (16). Recent work explores VLMs as zero-shot reward models (28) and addresses reward model uncertainty (8). LLM-as-a-judge approaches (39) provide scalable evaluation by using language models to assess output quality, which we adapt for semantic answer matching. Constitutional AI (13) demonstrates that AI-generated feedback can effectively replace human annotations. However, these reward formulations focus on absolute quality assessment rather than relative utility measurement. Our gain-based reward uniquely measures the *difference* in performance with and without captions, directly capturing whether visual descriptions provide task-relevant information that improves reasoning outcomes.

## 3 METHODOLOGY

We propose Caption as Reward (CaR), a reinforcement learning framework that optimizes visual descriptions based on their utility for downstream reasoning tasks. Unlike traditional caption evaluation that relies on linguistic metrics (BLEU, ROUGE, CLIPScore), CaR directly measures whether a caption improves reasoning accuracy. This section first analyzes the perception bottleneck in visual reasoning (§3.1), then presents our gain-based reward mechanism (§3.2), and finally describes the training procedure (§3.3).

### 3.1 PROBLEM ANALYSIS AND MOTIVATION

To understand failure modes in visual reasoning, we manually analyzed 1,200 multimodal math and science problems from MathVista and ScienceQA. For each failed case, we examined whether the model could solve a text-only version of the same problem when provided with an accurate visual description. This methodology isolates perception errors from reasoning errors: if the model succeeds with accurate descriptions but fails with its own visual understanding, the bottleneck lies in perception rather than reasoning.

Among the failures, 48.2% stemmed from missing or incorrect perceptual details in visual descriptions (e.g., misreading numbers, ignoring spatial relationships), 22.3% from reasoning errors (e.g., incorrect formula application, logical mistakes), and 29.4% from ambiguous cases where both factors contributed. After filtering ambiguous cases, perception-related errors comprised 62.1% of clear failures, as illustrated in Figure 1.

This analysis reveals a fundamental mismatch between how captions are evaluated and how they should be evaluated. Traditional metrics reward descriptive fluency—a caption like "a colorful chart with multiple bars" may score highly on linguistic quality while omitting the specific values needed for accurate reasoning. We argue that caption quality should be measured by reasoning utility: does the caption contain the information necessary to answer the question correctly?

## 3.2 CAPTION AS REWARD FRAMEWORK

Based on our analysis, we propose to evaluate captions through their downstream impact on reasoning performance. The key insight is that an effective caption should improve task accuracy when the model struggles with direct visual understanding, while preserving accuracy when the model already succeeds.

Given an image $I$, question $Q$, and ground-truth answer $A^*$, we consider two inference modes:

- **Direct reasoning**: The model processes the image and question jointly to produce an answer $A_{\text{direct}} \sim P_\theta(A \mid I, Q)$. This represents the standard VLM inference pipeline where visual perception is implicit.
- **Caption-enhanced reasoning**: The model first generates an explicit visual description $C \sim P_\theta(C \mid I, Q)$, then reasons over both the caption and image to produce $A_{\text{caption}} \sim P_\theta(A \mid I, C, Q)$. The caption serves as an intermediate representation that makes visual understanding explicit and verifiable.

Note that in caption-enhanced reasoning, the model still has access to the original image $I$, allowing it to verify or refine the caption during reasoning. This differs from purely text-based approaches that discard visual information after caption generation.

We define caption reward based on the performance difference between the two inference modes:

$$R(C \mid I, Q) = \begin{cases} 1.0, & \text{if } A_{\text{caption}} = A^* \wedge A_{\text{direct}} \neq A^* \quad \text{(fixes error)} \\ 0.7, & \text{if } A_{\text{caption}} = A^* \wedge A_{\text{direct}} = A^* \quad \text{(maintains accuracy)} \\ 0.2, & \text{if } A_{\text{caption}} \neq A^* \wedge A_{\text{direct}} \neq A^* \quad \text{(both fail)} \\ 0.0, & \text{if } A_{\text{caption}} \neq A^* \wedge A_{\text{direct}} = A^* \quad \text{(degrades performance)} \end{cases} \tag{1}$$

The reward values reflect our design priorities. Captions that fix reasoning errors ($R = 1.0$) provide the strongest positive signal, as they demonstrate clear utility for the task. Captions that maintain correct predictions ($R = 0.7$) are rewarded but less strongly, since they do not demonstrate improvement over direct reasoning. Captions where both modes fail ($R = 0.2$) receive minimal positive reward to encourage exploration. Critically, captions that degrade correct predictions ($R = 0.0$) are penalized, discouraging the model from generating descriptions that introduce errors or hallucinations.

Our approach differs fundamentally from Visionary-R1's RLAIF reward (37). Visionary-R1 uses an external VLM to judge whether a caption is "informative and accurate" based on absolute quality assessment. This approach has two limitations: (1) it relies on the external model's subjective judgment of caption quality, which may not align with task requirements; (2) it does not consider whether the caption actually helps reasoning. In contrast, CaR measures caption quality through a controlled experiment—comparing performance with and without the caption. A caption is rewarded only if it demonstrably improves reasoning outcomes, providing a more direct and task-aligned training signal.

The total training reward combines three components:

$$R_{\text{total}} = w_{\text{acc}} \cdot R_{\text{acc}} + w_{\text{format}} \cdot R_{\text{format}} + w_{\text{caption}} \cdot R(C \mid I, Q) \tag{2}$$

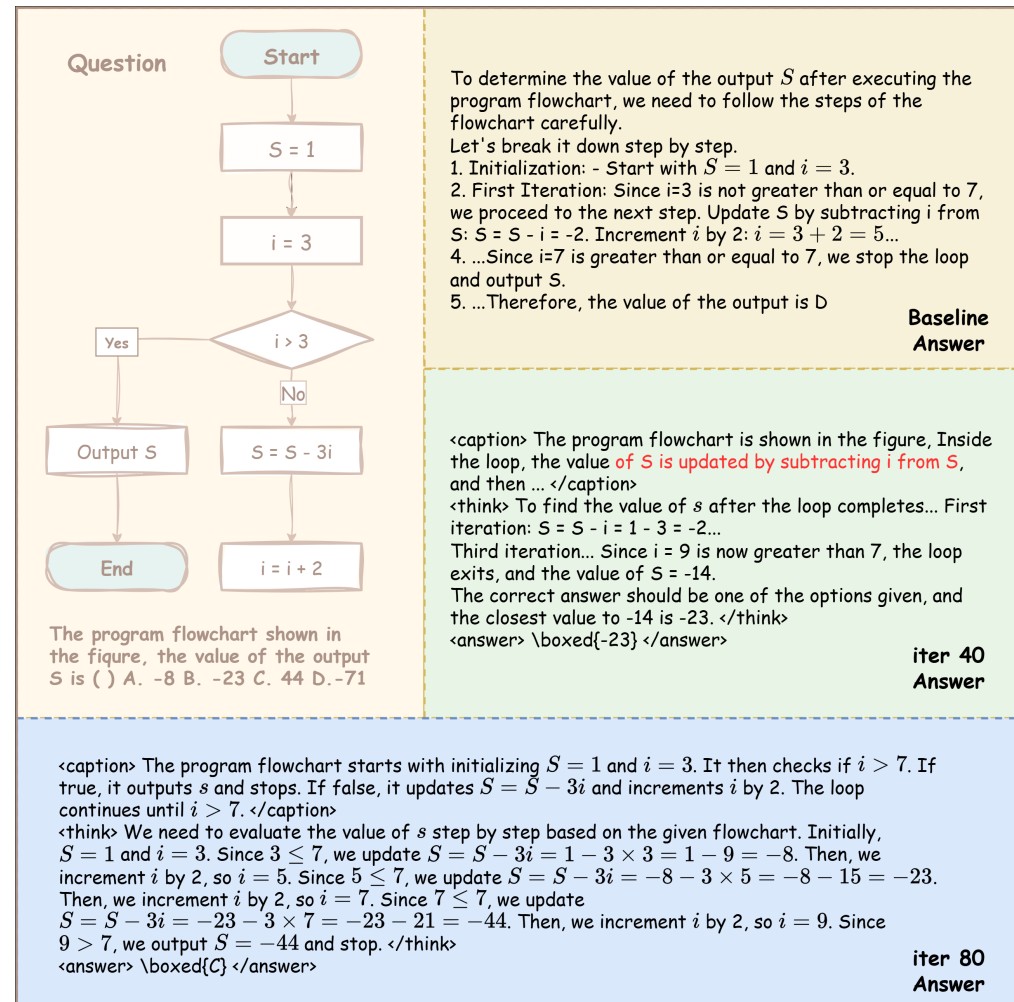

Figure 2: Evolution of caption quality during CaR training on a flowchart reasoning problem. **Baseline** (yellow): Direct reasoning without explicit caption produces incorrect answer (D). **Iter 40** (green): Early-stage caption captures partial information ("S is updated by subtracting i from S") but misses initialization details, leading to approximate answer (-23). **Iter 80** (blue): Fully-trained model generates complete caption with precise variable initialization ($S = 1$, $i = 3$), loop condition ($i > 7$), and update rules ($S = S - 3i$, $i = i + 2$), enabling correct step-by-step reasoning and final answer (C: -44).

where $R_{\text{acc}} \in \{0, 1\}$ indicates final answer correctness against ground truth, $R_{\text{format}} \in \{0, 1\}$ ensures proper output structure (caption, reasoning, answer sections), and $R(C \mid I, Q)$ is the gain-based caption reward from Equation 1. We set weights $(w_{\text{acc}}, w_{\text{format}}, w_{\text{caption}}) = (1.0, 0.1, 1.0)$, giving equal importance to final accuracy and caption utility while using a lower format weight since format compliance is typically achieved early in training.

## 3.3 TRAINING IMPLEMENTATION

We integrate caption rewards with Group Relative Policy Optimization (GRPO) (30), which provides stable training without requiring a separate critic network.

For each training sample, we generate $n = 8$ caption candidates using temperature sampling ($\tau = 0.7$). Computing absolute rewards can lead to high variance across samples of different difficulty. Following GRPO, we normalize rewards within each group:

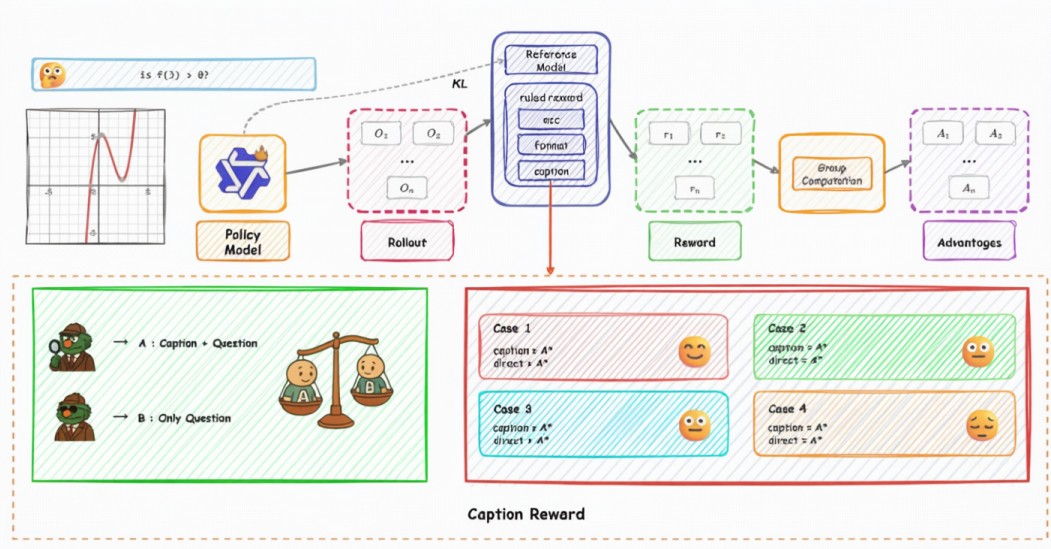

Figure 3: CaR training pipeline. For each image-question pair, the policy model generates multiple caption candidates. Each candidate is evaluated through both direct reasoning (without caption) and caption-enhanced reasoning to compute performance gains. The gain-based rewards are normalized within each group, and GRPO updates optimize the policy to favor captions with higher reasoning utility.

$$\hat{R}_i = \frac{R_i - \mu_g}{\sigma_g + \epsilon} \tag{3}$$

where $\mu_g$ and $\sigma_g$ are the mean and standard deviation of rewards within the group, and $\epsilon = 10^{-8}$ prevents division by zero. This normalization ensures that the model receives meaningful gradient signals regardless of sample difficulty.

The policy is updated using clipped importance sampling to prevent large deviations from the reference policy:

$$\mathcal{L}(\theta) = -\mathbb{E}_{(I,Q,C)\sim\mathcal{D}} \left[ \min\left( \rho_\theta \cdot \hat{R}, \ \text{clip}(\rho_\theta, 1 - \varepsilon, 1 + \varepsilon) \cdot \hat{R} \right) \right] \tag{4}$$

where $\rho_\theta = \pi_\theta(C \mid I, Q)/\pi_{\text{ref}}(C \mid I, Q)$ is the importance ratio between the current policy and reference policy, and $\varepsilon = 0.2$ is the clipping threshold. The reference policy is updated periodically (every 100 steps) to track the evolving policy while maintaining training stability.

A critical design choice is how to assess answer correctness for computing $R_{\text{acc}}$ and the gain-based reward. Using the training model itself would create a circular dependency where the model judges its own outputs. Instead, we employ an external evaluator—either GPT-4o-mini or Qwen2.5-7B-Instruct—to determine semantic equivalence between predicted and ground-truth answers.

The evaluator receives the question, ground-truth answer, and predicted answer, then outputs a binary correctness judgment. This design has two advantages: (1) it provides unbiased assessment independent of the training model's beliefs; (2) it handles semantic equivalence (e.g., "0.5" vs "1/2" vs "half") better than exact string matching. We found that GPT-4o-mini provides more consistent judgments with lower variance across similar samples, so we use it as the default evaluator.

To ensure clean separation between perception and reasoning phases, we enforce a structured output format with explicit XML-style tags:

```
<caption>...</caption> <think>...</think> <answer>...</answer>
```

Table 1: Comprehensive performance comparison across visual reasoning benchmarks. CaR denotes our Caption as Reward method.

| | MMK12 | MathVista | MathVision | MathVerse | DynaMath | WeMath | LogicVista | OlympiadBench | Avg |
|---|---|---|---|---|---|---|---|---|---|
| *Qwen2.5-VL-3B* | | | | | | | | | |
| GRPO | 57.6 | 66.1 | 25.3 | 30.1 | 14.4 | 28.8 | 42.1 | 8.8 | 34.2 |
| Ours | 60.1 | 68.1 | 26.2 | 31.1 | 15.0 | 30.1 | 41.4 | 8.8 | 35.1 |
| *InternVL2.5-4B* | | | | | | | | | |
| GRPO | 58.8 | 67.5 | 26.0 | 30.8 | 14.9 | 29.4 | 43.4 | 8.6 | 34.9 |
| Ours | 61.6 | 69.7 | 27.0 | 31.9 | 15.4 | 30.3 | 43.5 | 8.6 | 36.0 |
| *Qwen2.5-VL-7B* | | | | | | | | | |
| GRPO | 61.2 | 71.5 | 27.3 | 32.5 | 15.5 | 31.0 | 45.8 | 9.0 | 36.7 |
| Ours | 63.8 | 74.3 | 28.5 | 33.9 | 16.2 | 32.3 | 46.4 | 9.2 | 38.1 |
| *Baseline Comparisons (3B)* | | | | | | | | | |
| Qwen2.5-VL-3B | 41.1 | 61.2 | 21.9 | 31.2 | 13.2 | 22.9 | 40.0 | 6.8 | 29.8 |
| Visionary-R1 | 45.3 | 69.4 | 24.7 | 33.0 | 13.8 | 28.0 | 41.6 | 7.8 | 33.0 |
| TBAC-VLR1 | 47.2 | 64.8 | 25.0 | 34.5 | 17.7 | 32.4 | 40.8 | 8.3 | 33.8 |
| VLAA-Thinker-3B | 43.2 | 61.0 | 24.4 | 36.4 | 18.2 | 33.8 | 38.5 | 7.9 | 32.9 |

The `<caption>` section contains the visual description, `<think>` contains the reasoning process, and `<answer>` contains the final response. This structure allows us to extract and evaluate each component independently during reward computation. Format compliance is checked automatically, and $R_{\text{format}} = 1$ only when all three sections are present and properly formatted.

We sample 30K visual question-answer pairs from two existing datasets: MM-Eureka (17) (mathematical reasoning) and VirL39K (33) (logical reasoning). Importantly, we use only the question-answer pairs without any chain-of-thought annotations or human-written captions. The model must learn to generate useful captions and reasoning chains purely through reinforcement learning signals. This demonstrates that CaR can improve visual reasoning without expensive annotation efforts.

## 4 EXPERIMENTS

### 4.1 EXPERIMENTAL SETUP

We train models using two established multimodal reasoning datasets: MM-Eureka (17), focusing on mathematical problem-solving, and VirL39K (33), targeting logical inference tasks. We apply stratified sampling based on difficulty levels determined through eight inference runs with Qwen2.5-VL-7B (34), creating a balanced training set of 30K samples with a 90%/10% mix of medium and high-difficulty items. For evaluation, we assess performance across eight challenging visual reasoning benchmarks: MathVista (23), MathVision, MathVerse, DynaMath, WeMath, LogicVista, MMK12-EVAL, and OlympiadBench. All experiments use Qwen2.5-VL models at 3B and 7B parameter scales with GRPO (group size $n = 8$, learning rate $5 \times 10^{-7}$, temperature 0.9) for two epochs. The composite reward weights are set to $(1.0, 0.1, 1.0)$ for accuracy, format, and caption components respectively, with GPT-4o-mini or Qwen2.5-7B-Instruct serving as external evaluators. Training is conducted using the VeRL framework on NVIDIA A100 GPUs.

### 4.2 MAIN RESULTS

Table 1 presents comprehensive results across all evaluation benchmarks. CaR demonstrates consistent improvements over baseline methods across different model scales. The 3B model achieves 35.1% average accuracy, improving +0.9 points over the GRPO baseline (34.2%) and +5.3 points over the 3B-Instruct baseline (29.8%).

Notably, CaR shows particularly strong improvements on individual benchmarks. On MMK12-EVAL, the 3B model improves from 57.6% to 60.1% (+2.5 points), and on MathVista from 66.1% to 68.1% (+2.0 points). The 7B model exhibits even larger gains, improving from 36.7% to 38.1% average (+1.4 points), with notable improvements on MathVista (71.5% to 74.3%, +2.8 points) and MMK12 (61.2% to 63.8%, +2.6 points).

Table 2: Performance on MME-RealWorld benchmark for visual perception evaluation. Results show improvements in both reasoning and perception capabilities.

| Model | Reasoning | | | | | Perception | | | | | Avg |
|---|---|---|---|---|---|---|---|---|---|---|---|
| | Monitor | Auto Drive | OCR | Diagram | Remote | Monitor | Auto Drive | OCR | Diagram | Remote | |
| Qwen2.5-VL-3B-Instruct | 22.5 | 30.0 | 57.8 | 46.6 | 0.0 | 31.6 | 35.7 | 70.3 | 40.1 | 22.6 | 35.7 |
| Ours | 21.9 | 28.2 | 60.8 | 52.2 | 0.0 | 33.1 | 37.2 | 78.4 | 71.5 | 26.9 | 41.0 |

Table 3: Impact of different external evaluators on caption reward quality.

| | MMK12 | MathVista | MathVision | MathVerse | DynaMath | WeMath | LogicVista | OlympiadBench | Avg |
|---|---|---|---|---|---|---|---|---|---|
| Qwen2.5-7B-Instruct | 50.4 | 65.1 | 22.0 | 30.5 | 12.8 | 25.2 | 38.9 | 7.5 | 31.6 |
| gpt-4o-mini | 49.8 | 66.3 | 22.0 | 31.6 | 12.3 | 27.2 | 43.4 | 7.9 | 32.8 |

To validate that CaR's effectiveness extends beyond a single model family, we also evaluate on InternVL2.5-4B (4). Results show consistent improvements (+1.1 points average), demonstrating that the gain-based reward mechanism transfers across different VLM architectures. This cross-architecture consistency suggests that CaR captures fundamental properties of useful visual descriptions rather than architecture-specific patterns.

Performance improvements vary across benchmarks, with the largest gains on tasks requiring precise visual extraction. MathVista shows consistent improvement across all model scales, while DynaMath and OlympiadBench—which involve complex multi-step reasoning—show more modest gains. This pattern suggests that CaR primarily improves the perception component of visual reasoning, with downstream reasoning benefiting indirectly from more accurate visual information.

Compared to recent competitive baselines, our 3B model (35.1%) outperforms Visionary-R1 (33.0%), TBAC-VLR1 (33.8%), and VLAA-Thinker-3B (32.9%) on average accuracy across all benchmarks.

## 4.3 VISUAL PERCEPTION ANALYSIS

To evaluate CaR's impact on visual perception specifically, we conduct additional experiments on MME-RealWorld, a benchmark designed for fine-grained visual understanding across diverse real-world scenarios. Table 2 shows detailed results across reasoning and perception tasks.

CaR demonstrates substantial improvements in visual perception capabilities, achieving 41.0% average performance compared to 35.7% for the baseline (+5.3 points). The most significant gains occur in Perception-Diagram (+31.4 points: 71.5 vs 40.1) and Perception-OCR (+8.1 points: 78.4 vs 70.3). These results confirm that CaR's gain-based reward mechanism effectively enhances the model's ability to extract task-relevant visual information.

The performance gains are not uniform across all perception categories. Diagram understanding shows the largest improvement, likely because mathematical diagrams require precise extraction of numerical values, geometric relationships, and labels—exactly the type of information that CaR's gain-based reward incentivizes. OCR tasks also benefit substantially, as accurate text recognition is often critical for correct reasoning. In contrast, autonomous driving and remote sensing tasks show more modest improvements, possibly because these domains require different types of visual understanding that are less directly captured by our question-answering training format.

Interestingly, some reasoning subtasks show slight decreases (e.g., Monitor: 22.5 to 21.9, Auto Drive: 30.0 to 28.2) while corresponding perception tasks improve. This suggests that CaR may shift the model's focus toward detailed visual extraction at a small cost to high-level reasoning in certain domains. However, the overall average improvement (+5.3 points) indicates that the perception gains substantially outweigh any reasoning trade-offs, validating CaR's design philosophy of prioritizing accurate visual understanding as a foundation for reasoning.

## 4.4 ANALYSIS

We investigate the impact of different evaluation models for caption reward calculation (Table 3). GPT-4o-mini yields superior performance (32.8% average) compared to Qwen2.5-7B-Instruct (31.6%), with a +1.2 points difference. This suggests that more capable evaluators provide cleaner training signals with reduced evaluation noise, highlighting the importance of evaluator quality in RL-based training pipelines.

Analysis of 100 samples where CaR succeeded but baselines failed reveals consistent improvement patterns. First, CaR generates specific numerical values and spatial relationships rather than approximate descriptions—for example, describing "a triangle with sides 3, 4, and 5 units" instead of "a right triangle." Second, models learn to prioritize information directly relevant to the question, emphasizing geometric relationships in math problems over aesthetic details like colors or backgrounds. Third, the resulting descriptions follow structured patterns that systematically cover different image regions, reducing the likelihood of missing critical visual elements.

Regarding computational cost, CaR introduces additional inference passes for caption evaluation. However, several optimizations minimize overhead: (1) the question-only inference results can be precomputed before training, eliminating network I/O during the training loop; (2) during rollout, partially generated captions that share common prefixes can reuse cached computations; (3) the caption evaluation and importance sampling can be computed in parallel, synchronizing only at the final loss computation to avoid blocking the training pipeline. With these optimizations, CaR remains competitive with other RL-based methods in terms of training efficiency.

## 5 DISCUSSION AND LIMITATIONS

Our analysis suggests several factors contribute to CaR's effectiveness. By directly optimizing for task utility rather than proxy metrics like linguistic similarity, CaR creates a tight feedback loop between perception quality and reasoning accuracy. The gain-based reward also naturally establishes an implicit curriculum where the model learns to fix obvious perception errors before gradually improving on subtler cases—this organic difficulty progression may explain why CaR scales better with data than SFT. Additionally, separate evaluation of direct and caption-enhanced reasoning provides disentangled learning signals that clarify what visual information is missing, helping the model identify task-relevant visual features rather than memorizing caption patterns.

Unlike linguistic metrics (BLEU, ROUGE) that measure surface-level similarity, CaR's gain-based reward directly captures reasoning utility. This fundamental difference explains why CaR outperforms caption-based methods that rely on linguistic optimization. Compared to self-consistency approaches that check reasoning agreement, CaR provides stronger supervision by requiring captions to demonstrably improve task performance. The use of external evaluators also avoids the self-reinforcement bias that can occur when models judge their own outputs.

Our experiments demonstrate that CaR generalizes across different model scales (3B, 4B, 7B) and task types (mathematical reasoning, logical inference, visual perception). The consistent improvements suggest that the gain-based reward captures fundamental aspects of visual description quality that transfer across domains. However, the degree of improvement varies by task: CaR shows larger gains on tasks requiring precise numerical extraction (e.g., diagram understanding) compared to tasks dominated by spatial reasoning.

While CaR demonstrates strong results, several limitations warrant discussion. The approximately $3\times$ computational overhead compared to SFT may limit adoption for resource-constrained settings; future work should explore more efficient reward computation strategies such as caching evaluator responses or using lightweight reward models. We evaluated CaR only on Qwen2.5-VL (34) and InternVL (4) models, and testing on more diverse architectures (LLaVA (20; 21), BLIP (15), MiniGPT-4 (40)) would strengthen claims about generalizability. Current experiments focus on question-answering tasks, and extending CaR to other modalities (video, audio) and tasks (generation, editing) remains unexplored. Our reward weights and thresholds were determined through limited grid search, and more principled approaches using multi-objective optimization or learned reward functions could improve performance.

CaR's improved visual perception could enable more reliable VLM deployments in education, accessibility, and scientific research. For educational applications, accurate visual understanding is critical for tutoring systems that explain diagrams and figures. In accessibility contexts, better caption generation could improve screen reader experiences for visually impaired users. However, enhanced visual understanding also raises concerns about potential misuse for surveillance or generating misleading content. We recommend careful deployment with appropriate safeguards and regular auditing of model outputs to detect potential misuse.

## 6 CONCLUSION

We introduced Caption as Reward (CaR), a reinforcement learning framework that optimizes visual descriptions based on their utility for downstream reasoning tasks. CaR addresses a fundamental limitation in vision-language model training: the disconnect between linguistic caption quality and reasoning performance. By evaluating captions through performance gains rather than linguistic similarity metrics, CaR provides a principled training signal that directly aligns visual understanding with task requirements.

Our experimental results demonstrate CaR's effectiveness across eight challenging visual reasoning benchmarks. The 3B model achieves 35.1% average accuracy, outperforming the instruction-tuned baseline (29.8%, +5.3 points) and the GRPO baseline (34.2%, +0.9 points). The 7B model shows consistent improvements from 36.7% to 38.1% (+1.4 points), demonstrating scalability across model sizes. Cross-architecture validation on InternVL2.5-4B (+1.1 points) confirms that CaR's benefits transfer beyond a single model family.

The key insight underlying CaR is that visual description quality should be measured by reasoning utility rather than descriptive fluency. This performance-centric approach enables models to learn what visual information matters for specific tasks, resulting in more precise and task-relevant visual understanding. Additional evaluation on MME-RealWorld confirms enhanced perception capabilities, with particularly strong improvements in diagram understanding (+31.4 points) and OCR tasks (+8.1 points).

Current limitations include evaluation on two model families (Qwen2.5-VL and InternVL) and reliance on external evaluators for reward computation. Future work will explore broader architectural validation across diverse VLM designs, extension to video and audio modalities, and application to generative tasks beyond question answering.

CaR opens new directions for task-adaptive multimodal learning by demonstrating that performance-based optimization can effectively bridge the perception-reasoning gap in vision-language models without requiring human annotations or auxiliary reward models. This approach may generalize to other domains where proxy metrics (e.g., perplexity, BLEU) poorly correlate with downstream task performance, suggesting a broader paradigm of utility-based optimization for foundation models.

## ETHICS STATEMENT

CaR targets educational and scientific reasoning workloads by strengthening factual visual under-standing. Although stronger perception could be misused to generate misleading analyses, the method reduces hallucinated descriptions and relies only on public datasets, which we acknowledge may carry existing societal biases. We adhere to the ICLR Code of Ethics and confirm that our work complies with all ethical guidelines. All authors of this work have read and commit to adhering to the ICLR Code of Ethics.

## REPRODUCIBILITY STATEMENT

We train on the publicly available MM-Eureka and VirL39K corpora using Qwen2.5-VL models (3B/7B) and InternVL2.5-4B. All hyperparameters (GRPO with $n$=8, learning rate $5 \times 10^{-7}$, temperature 0.9, two epochs, reward weights $(1.0, 0.1, 1.0)$) and evaluator choices (GPT-4o-mini or Qwen2.5-7B-Instruct) are described in Section 4.1. Code, data splits, and checkpoints will be released upon acceptance.

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

## A  TECHNICAL APPENDICES AND SUPPLEMENTARY MATERIAL

### A.1  POLICY MODEL PROMPT

To ensure the model generates structured visual descriptions before reasoning, we include additional instructions in the system prompt to guide the policy model in generating the corresponding output. Using this prompt, the model will insert the corresponding image description labeled as ¡caption¿ before the thinking process, additional to the existing ¡think¿ and ¡answer¿.

---

**Policy Model Prompt**

A conversation between User and Assistant. The user asks a question, and the Assistant solves it. The assistant begins the response with a concise, image-grounded caption enclosed in `<caption>` `</caption>`, then thinks about the reasoning process in the mind and provides the answer. The reasoning process and answer are enclosed within `<think>` `</think>` and `<answer>` `</answer>` tags, respectively; i.e., `<caption>` caption here `</caption>` `<think>` reasoning process here `</think>` `<answer>` answer here `</answer>`.

---

### A.2  GAIN-BASED REWARD IMPLEMENTATION

Algorithm 1 summarizes the reward computation pipeline referenced in Section 3.2.

---

**Algorithm 1** Gain-Based Reward Computation

---

**Require:** Image $I$, Question $Q$, Generated Caption $C$, Ground Truth $A^*$, Vision-Language Model $\mathcal{M}$, External Evaluator $\mathcal{E}$
**Ensure:** Gain-based reward $R(C|I,Q)$
1: $A_{\text{direct}} \leftarrow \mathcal{M}(I,Q)$ {Direct reasoning}
2: $A_{\text{caption}} \leftarrow \mathcal{M}(I,Q,C)$ {Caption-enhanced reasoning}
3: $\text{correct}_{\text{direct}} \leftarrow \mathcal{E}.\text{SemanticMatch}(A_{\text{direct}}, A^*)$
4: $\text{correct}_{\text{caption}} \leftarrow \mathcal{E}.\text{SemanticMatch}(A_{\text{caption}}, A^*)$
5: **if** $\text{correct}_{\text{caption}}$ **and** $\neg\text{correct}_{\text{direct}}$ **then**
6:     $R(C|I,Q) \leftarrow 1.0$ {Caption fixes an error}
7: **else if** $\text{correct}_{\text{caption}}$ **and** $\text{correct}_{\text{direct}}$ **then**
8:     $R(C|I,Q) \leftarrow 0.7$ {Caption confirms success}
9: **else if** $\neg\text{correct}_{\text{caption}}$ **and** $\neg\text{correct}_{\text{direct}}$ **then**
10:     $R(C|I,Q) \leftarrow 0.2$ {Both attempts fail}
11: **else**
12:     $R(C|I,Q) \leftarrow 0.0$ {Caption harms accuracy}
13: **end if**
14: **return** $R(C|I,Q)$

---

## A.3 TRAINING DATA SOURCES

Our training dataset combines two established open-source multimodal reasoning corpora: MM-Eureka (17) and VirL39K (33), covering diverse visual reasoning tasks including mathematical problem-solving and spatial reasoning. We apply stratified sampling based on difficulty levels to ensure effective training data quality, following the data construction methodology described in Section 3.3.

## A.4 ADDITIONAL EXPERIMENTAL RESULTS

**Data Scaling Analysis** We analyze the impact of training data size on CaR performance. Table 4 shows results with 10K, 20K, and 30K training samples on the 3B model. CaR demonstrates consistent improvements over GRPO across all data scales: 10K (+0.6 points), 20K (+1.2 points), and 30K (+1.3 points). This scaling trend indicates that CaR effectively leverages additional training data to enhance visual reasoning capabilities.

Table 4: Impact of training data size on CaR performance (Qwen2.5-VL-3B).

| Data Size | Method | MMK12 | MathVista | MathVision | MathVerse | DynaMath | WeMath | LogicVista | OlympiadBench | Avg |
|---|---|---|---|---|---|---|---|---|---|---|
| 10K | SFT | 29.6 | 54.7 | 18.8 | 8.1 | 9.7 | 16.2 | 34.9 | 6.0 | 22.2 |
| 10K | GRPO | 48.5 | 64.1 | 22.6 | 32.0 | 12.5 | 27.0 | 41.2 | 8.0 | 32.0 |
| 10K | CaR | 49.8 | 66.3 | 22.0 | 31.6 | 12.3 | 27.7 | 43.4 | 7.9 | 32.6 |
| 20K | SFT | 33.7 | 54.0 | 18.6 | 9.1 | 9.2 | 18.2 | 34.0 | 6.0 | 22.9 |
| 20K | GRPO | 51.2 | 64.8 | 23.4 | 28.5 | 13.1 | 27.5 | 41.8 | 8.2 | 32.3 |
| 20K | CaR | 53.5 | 66.9 | 24.2 | 29.4 | 13.6 | 28.4 | 43.1 | 8.5 | 33.5 |
| 30K | SFT | 38.5 | 61.5 | 19.1 | 23.6 | 4.3 | 17.9 | 36.7 | 7.1 | 26.1 |
| 30K | GRPO | 54.8 | 63.9 | 24.5 | 29.2 | 13.9 | 27.9 | 40.8 | 8.5 | 32.9 |
| 30K | CaR | 57.6 | 66.1 | 25.3 | 30.1 | 14.4 | 28.8 | 42.1 | 8.8 | 34.2 |

**Hyper-parameter Analysis** We analyze the sensitivity of CaR to the caption reward weight $w_{caption}$ in the composite reward structure. For the accuracy and format rewards, we follow standard GRPO settings with equal weights. Table 5 shows results across all benchmarks for different caption reward weights. We observe that $w_{caption} = 0.1$ achieves the best overall performance (35.1% average), while higher weights lead to slight degradation. To maintain training stability and prevent caption reward from dominating the optimization objective, we recommend using $w_{caption} = 0.1$ in practice.

Table 5: Hyper-parameter analysis for caption reward weight $w_{caption}$ (Qwen2.5-VL-3B).

| Weight $w_{caption}$ | MMK12 | MathVista | MathVision | MathVerse | DynaMath | WeMath | LogicVista | OlympiadBench | Avg |
|---|---|---|---|---|---|---|---|---|---|
| 0.1 | 60.1 | 68.1 | 26.2 | 31.1 | 15.0 | 30.1 | 41.4 | 8.8 | 35.1 |
| 0.5 | 57.0 | 64.7 | 24.6 | 29.5 | 14.1 | 28.3 | 39.1 | 8.3 | 33.2 |
| 1.0 | 55.3 | 62.9 | 24.2 | 28.7 | 13.8 | 27.7 | 38.3 | 8.1 | 32.4 |

## A.5 VISUALIZATION OF THE CAR OUTPUT

We present four representative examples demonstrating CaR's structured output format with caption, thinking, and answer components.

**Diagram Reasoning**

**Q:** How many predators does the mouse have?
Options: 4, 1, 0, 2
**A:** 1

**CaR Output**

**\<caption\>** A food web diagram showing ecological relationships. Primary producers (berries, grass, trees) are at the bottom. Primary consumers include insects, deer, and mice. Secondary consumers are skunks, wolverine, and weasel. Tertiary consumers at top are arctic fox, great horned owl, and hawks. Blue arrows show energy flow from prey to predator. **\</caption\>**

**\<think\>** Looking at the food web, I need to find which animals eat mice. Following the arrows FROM mice, I can see only one arrow pointing to weasel. The weasel is the only predator that consumes mice in this diagram. **\</think\>**

**\<answer\>** 1 **\</answer\>**

**Table Understanding**

| Refrigerator | Initial temperature (°C) | Final temperature (°C) |
|---|---|---|
| Refrigerator with its door open | 3.9 | 11.4 |
| Refrigerator with its door closed | 3.9 | 5.4 |

**Q:** During this time, thermal energy was transferred from () to ().
A. each refrigerator...the surroundings
B. the surroundings...each refrigerator
**A:** B

**CaR Output**

**\<caption\>** A table showing temperature changes in two refrigerators. Row 1: Refrigerator with door open, initial temp 3.9°C, final temp 11.4°C. Row 2: Refrigerator with door closed, initial temp 3.9°C, final temp 5.4°C. Both refrigerators increased in temperature over 10 minutes. **\</caption\>**

**\<think\>** Both refrigerators started at 3.9°C and ended at higher temperatures (11.4°C and 5.4°C). Since temperature increased, thermal energy must have flowed INTO the refrigerators FROM the warmer surroundings, not the other way around. **\</think\>**

**\<answer\>** B **\</answer\>**

## Chart Analysis

**Adults ages 25 to 34 are most likely to be cohabiting**

*% who are ____, by age group*

■ Cohabiting ■ Married ■ Divorced or ■ Never ▪ Separated
                        widowed        married

| Age | | | | | |
|---|---|---|---|---|---|
| 18-24 | 9 | 7 | 1 | 82 | 2 |
| 25-34 | 14 | 41 | 3 | 38 | 3 |
| 35-49 | 8 | 63 | 10 | 15 | 4 |
| 50+ | 4 | 61 | 25 | 8 | 3 |

Note: All individuals who were cohabiting are included only in the cohabiting category, regardless of marital status. "Separated" includes those who were married, but their spouse was absent.
Source: Pew Research Center analysis of 2016 Current Population Survey, Annual Social and Economic Supplement (IPUMS).

**PEW RESEARCH CENTER**

**Q:** How many colors are in the bar?
**A:** 5

## CaR Output

**\<caption\>** A horizontal stacked bar chart titled "Adults ages 25 to 34 are most likely to be cohabiting." Shows percentages by age group (18-24, 25-34, 35-49, 50+) with 5 categories: Cohabiting (dark green), Married (light green), Divorced/widowed (beige), Never married (olive), and Separated (gray). **\</caption\>**

**\<think\>** Counting the distinct colors in the legend: 1) Cohabiting (dark green), 2) Married (light green), 3) Divorced or widowed (beige), 4) Never married (olive/brown), 5) Separated (gray). Total: 5 colors. **\</think\>**

**\<answer\>** 5 **\</answer\>**

## Geometry Problem

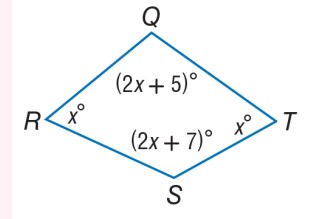

**Q:** Find m∠S.
**A:** 123°

## CaR Output

**\<caption\>** A quadrilateral QRST (kite shape) with vertices Q at top, R at left, S at bottom, T at right. Angle at R is x°, angle at T is x°. Angle at Q is $(2x+5)°$, angle at S is $(2x+7)°$. The shape appears symmetric about the vertical axis. **\</caption\>**

**\<think\>** Sum of angles in a quadrilateral = 360°. So: $x + x + (2x+5) + (2x+7) = 360$. Simplifying: $6x + 12 = 360$, thus $6x = 348$, $x = 58$. Therefore, $m\angle S = 2(58) + 7 = 116 + 7 = 123°$. **\</think\>**

**\<answer\>** 123° **\</answer\>**

