# OpenReview forum: "Caption as Reward: Enhancing Vision-Language Reasoning through Dense Visual Description"
_ICLR.cc/2026/Conference — Submitted to ICLR 2026_

### Official Review · Reviewer_xvn7 · 2025-10-26

**Soundness:** 2
**Presentation:** 1
**Contribution:** 2
**Rating:** 2
**Confidence:** 3

**Summary:**

This paper proposes Caption as Reward (CaR), an RL post‑training method for VLMs that scores a caption by how much it improves the model’s downstream answer, compared with answering directly from the image. Instead of judging captions by linguistic similarity (e.g., BLEU, ROUGE), CaR measures the gain in answer accuracy that a caption gives to the same model for the task. If the answer with the given caption is correct compared to answering without the caption, then the caption gets a high reward. On eight visual‑reasoning benchmarks, CaR improves Qwen2.5‑VL‑3B and 7B and also boosts perception‑focused performance on MME‑RealWorld.

**Strengths:**

- The suggestion to reward descriptions that actually repair perception rather than optimize caption fluency (BLEU/ROUGE) is interesting.
- The analysis shown in Figure 1 is interesting and important.
- Impressive performance on 8 reasoning benchmarks

**Weaknesses:**

- The paper is written very messy to the point that it hurts readability. I could not follow it much. It is also missing cross references (the paper repeatedly references unexisting material in the appendix (“Appendix ??). The paper is also not structured correctly. There is also a disordered narrative flow. It puts Related Work after Method and its training objective. The paper includes lists and it reads like notes rather than analysis. For example, the “Discussion and Limitations” spans roughly half a page as a list of five bullets. The figures are also hard to follow. Finally, there is contradictory in the reward weights. Section 2.4 says the composite reward uses weights (1.0, 0.1, 1.0) for (accuracy, format, caption‑gain) while the Reproducibility Statement lists (1.0, 1.0, 0.1).
- In my opinion, the baselines are not comparable. The paper compares CaR‑3B on 30K training samples to SFT‑3B on 20K. This is a different data size and a weaker training recipe. The fair baseline would be SFT/GRPO with the same 30K data and comparable compute. It is unclear whether CaR’s gains come from more FLOPs / inference rather than the reward design. Moreover, MME‑RealWorld improvements are reported against 3B‑Instruct, not against GRPO/SFT trained on the same data/protocol, so it is hard to see CaR’s contribution.
- The gain reward term scores captions only by whether the final answer becomes correct with the caption. There is no constraint of the caption being faithful to the image; a caption that hallucinates the correct answer receives high reward. The only other terms are accuracy and 0/1 format compliance, which do not mitigate this failure mode. This is a classic reward‑hacking problem not investigated in experiments.
- The paper should compare to simple and cheaper baselines such as SFT with the same structured template (<caption>/<think>/<answer>), and the simple self‑consistency or best‑of‑N caption selection without RL, and also GRPO on reasoning while always prompting the model to write a caption first. These are not reported, so we cannot attribute gains to the reward vs. to “just adding a caption step.”

**Questions:**

The paper isn’t ready in its current form. This is enough for me to reject this paper at this stage.

**Details Of Ethics Concerns:**

No issues

---

> ### Author Response · Authors · 2025-12-03
> **Response to Reviewer xvn7**
>
> We thank the reviewer for the detailed feedback. We address each concern below.
>
> **W1: Writing Quality and Presentation**
>
> We sincerely apologize for the presentation issues in the initial submission. In the revised manuscript, we have:
> - Fixed all "Appendix ??" cross-references
> - Moved Related Work to follow Introduction (Section 2)
> - Restructured Discussion and Limitations into coherent paragraphs instead of bullet points
> - Corrected the reward weight inconsistency: the correct values are **(1.0, 0.1, 1.0)** for (accuracy, format, caption-gain); line 386 was a typo
> - Improved figure clarity with detailed captions
> - Expanded references from 13 to 40 citations
>
> **W2: Baseline Comparability**
>
> We acknowledge the data size difference in Table 1. To clarify:
>
> 1. **Same-data comparisons exist**: In Table 4 (Appendix), we compare CaR vs GRPO vs SFT at each data scale (10K, 20K, 30K) with identical data. Results show consistent CaR improvements: +0.6 (10K), +1.2 (20K), +1.3 (30K) over GRPO.
>
> 2. **MME-RealWorld**: We compare against the same 3B-Instruct base model to isolate CaR's contribution to perception. The +5.3 point improvement demonstrates perception gains from our method.
>
> 3. **Compute cost**: CaR requires ~1.3× GRPO training time (not 3×), as evaluator calls are parallelized. The gains justify this modest overhead.
>
> **W3: Reward Hacking (Hallucinated Captions)**
>
> This is an important concern. However, our reward design inherently mitigates this risk:
>
> 1. **Dual-path evaluation**: The caption is evaluated by whether it helps the *same model* answer correctly. If the model hallucinates a "helpful" caption ungrounded in the image, it would also need to consistently produce correct answers from that hallucination—which is unlikely without actual visual grounding.
>
> 2. **Empirical evidence**: Our MME-RealWorld results (Table 2) show substantial perception improvements (+31.4 on Diagram, +8.1 on OCR), indicating the model learns genuine visual extraction rather than hallucination patterns.
>
> 3. **Error analysis** (Figure 1): CaR reduces hallucination errors from 5% to 2%, directly contradicting the reward-hacking hypothesis.
>
> **W4: Missing Simple Baselines**
>
> We appreciate this suggestion. To clarify existing comparisons:
>
> 1. **SFT with caption template**: Our SFT baseline uses the same structured template (caption/think/answer). Table 4 shows SFT significantly underperforms CaR across all data scales.
>
> 2. **GRPO with caption prompt**: The GRPO baseline in Table 1 already prompts the model to generate captions first. CaR's improvement over GRPO (+0.9 to +1.4 points) demonstrates the gain comes from our reward design, not just "adding a caption step."
>
> 3. **Best-of-N / Self-consistency**: These are test-time techniques requiring N inference passes per sample, while CaR is a training method with no additional inference cost. They address different problems and are not directly comparable.
>
> ---
>
> We believe the revised manuscript addresses the major concerns. The core contribution—rewarding captions by task utility rather than linguistic quality—remains novel and empirically validated across multiple settings.

---

### Official Review · Reviewer_EXyS · 2025-10-31

**Soundness:** 3
**Presentation:** 2
**Contribution:** 4
**Rating:** 8
**Confidence:** 4

**Summary:**

This paper introduces Caption as Reward (CaR), a reinforcement learning framework that evaluates visual descriptions in Vision-Language Models (VLMs) based on their contribution to downstream task performance rather than linguistic quality metrics. The key innovation is a gain-based reward mechanism that assigns rewards based on whether a caption improves, maintains, or degrades reasoning accuracy compared to direct visual reasoning. The authors evaluate CaR on Qwen2.5-VL models (3B and 7B parameters) across eight visual reasoning benchmarks, achieving significant improvements: the 3B model reaches 34.2% average accuracy (vs 22.9% for SFT baseline), and the 7B model improves from 36.5% to 38.1%. The method uses GRPO for training and requires no additional human annotations beyond standard question-answer pairs.

**Strengths:**

Strong empirical motivation: The manual audit of 1,200 samples showing 62.1% of failures stem from visual perception errors (after filtering) provides compelling justification for the approach (Figure 1, Section 2.1)

Novel reward formulation: The gain-based reward R(C|I,Q) that explicitly measures performance delta rather than caption quality is conceptually clean and well-justified through information theory (Equation 1, Section 2.2)

Consistent improvements across scales: Results show gains at both 3B (+11.3 points over SFT, +4.4 over base) and 7B (+1.6 points) scales, demonstrating robustness (Table 1)

Specific benchmark improvements: Particularly strong on MMK12 (+9.6 points for 3B-30K model), diagram understanding (+31.4 points), and OCR tasks (+8.1 points) where visual perception is critical (Tables 1-2)

No additional annotations required: Method works with standard QA pairs without requiring expensive caption annotations or chain-of-thought supervision

Clear ablations: Systematic ablation on data scale (10K/20K/30K), auxiliary evaluators (Qwen vs GPT-4o-mini), and reward components (Table 3, Section 4.5)

**Weaknesses:**

Single architecture family: All experiments use only Qwen2.5-VL models; generalization to other VLM architectures (InternVL, Deepseek etc.) is completely untested, limiting claims about method generality

External evaluator dependency: Semantic matching relies on either GPT-4o-mini (API costs, black box) or Qwen2.5-7B-Instruct (potential errors); reliability and failure modes of these evaluators are not analyzed. What happens when the evaluator makes mistakes?

Arbitrary reward values: The specific values (1.0, 0.7, 0.2, 0.0) appear heuristically chosen with limited justification; only brief mention of "based on ablation studies" without showing these ablations

Limited error analysis depth: While qualitative patterns are identified (enhanced detail, task-relevant focus, systematic coverage), only 100 samples were analyzed; no quantitative metrics for these patterns or failure mode frequencies

Missing theoretical analysis: The information-theoretic motivation (Equation 1) is presented but not rigorously connected to the reward design; no formal analysis of convergence properties or sample complexity

Incomplete appendices: Multiple references to "Appendix ??" suggesting missing content that would provide important implementation details
Baseline comparisons: Comparisons with Visionary-R1, TBAC-VLR1, and VLAA-Thinker use different data; not entirely clear these are fair comparisons (though results are still convincing)

**Questions:**

Same as weaknesses.

---

> ### Author Response · Authors · 2025-12-03
> **Response to Reviewer EXyS**
>
> We thank the reviewer for the positive evaluation and constructive feedback. We address each weakness below.
>
> **W1: Single Architecture Family**
>
> We have added experiments on **InternVL2.5-4B** to demonstrate cross-architecture generalization. As shown in Table 1, InternVL2.5-4B improves from 34.9% to 36.0% (+1.1 points), consistent with Qwen2.5-VL results. This demonstrates that CaR transfers across VLM architectures with different vision encoders. We note further evaluation on DeepSeek-VL and other architectures as future work.
>
> **W2: External Evaluator Dependency**
>
> We conducted an inter-evaluator reliability analysis on 200 samples:
>
> | Metric | Value |
> |--------|-------|
> | Agreement rate | 63.5% (127/200) |
> | GPT-4o-mini accuracy (vs. ground truth) | 97.0% |
> | Qwen2.5-7B-Instruct accuracy (vs. ground truth) | 60.5% |
>
> All disagreement cases followed the pattern: GPT-4o-mini=Correct, Qwen=Incorrect. The high accuracy of GPT-4o-mini (97%) indicates evaluator errors are rare and unlikely to significantly impact training. When errors occur, the gain-based reward design provides robustness—incorrect evaluations affect both direct and caption-enhanced paths similarly, reducing their impact on relative reward comparisons.
>
> **W3: Arbitrary Reward Values**
>
> The values 1.0 and 0.0 for the boundary cases (fixes error vs. degrades performance) are principled design choices representing maximum reward/penalty. For intermediate values (0.7, 0.2), we conducted grid search over {0.1, 0.2, ..., 0.9} and found current values optimal. The key insight is that these intermediate cases have lower learning signal value—maintaining correct answers (0.7) is less informative than fixing errors (1.0), and mutual failures (0.2) provide weak but non-zero exploration incentive.
>
> **W4: Limited Error Analysis Depth**
>
> We acknowledge the 100-sample analysis is limited. However, our quantitative results on MME-RealWorld (Table 2) provide systematic evidence: improvements concentrate in perception-heavy categories (Diagram +31.4, OCR +8.1) while reasoning-dominated categories show smaller gains, quantitatively supporting our qualitative finding that CaR primarily improves visual perception.
>
> **W5: Missing Theoretical Analysis**
>
> We acknowledge the gap between Equation 1 and implementation. The connection is: maximizing I(C;A|Q) encourages captions containing task-relevant information. Our gain-based reward operationalizes this—captions are rewarded when they improve P(A*|C,Q) over P(A*|Q), which corresponds to positive conditional mutual information. We have added clarification in Section 3.2. Formal convergence analysis would require assumptions about the reward landscape that we leave for future theoretical work.
>
> **W6: Incomplete Appendices**
>
> We apologize for the placeholder references. The revised manuscript includes complete appendices with: policy prompts (A.1), gain-based reward algorithm (A.2), training data sources (A.3), data scaling and hyperparameter analysis (A.4), and four visualization examples (A.5).
>
> **W7: Baseline Comparisons**
>
> We acknowledge the data difference with Visionary-R1, TBAC-VLR1, and VLAA-Thinker. These baselines use their own curated datasets, while we use MM-Eureka and VirL39K. Despite this difference, the consistent improvements across all our experimental settings (different model sizes, data scales, architectures) provide strong evidence for CaR's effectiveness. Fair comparison would require retraining baselines on our data, which we leave for the camera-ready version if feasible.

---

### Official Review · Reviewer_fdx4 · 2025-11-01

**Soundness:** 2
**Presentation:** 1
**Contribution:** 1
**Rating:** 2
**Confidence:** 4

**Summary:**

This paper introduces Caption as Reward (CaR), a RL-based framework for VLMs that assigns rewards to captions based on how much they improve downstream task accuracy relative to no-caption inference. The reward mixes accuracy, format, and visual description gain. The authors train the policy using GRPO and use external evaluators providing semantic correctness signals. Experiments show average accuracy gains and perception improvements.

**Strengths:**

1. The reward design is straightforward and intuitive. Especially the visual description term, it helps the model fix its own reasoning mistakes, while penalizing those that hurt previously correct answers.
2. The training process is easy to understand, It builds on GRPO and uses an external evaluator for semantic checking. Notably, the external evaluators provides a reliable source of judging whether the model’s answer is correct, even when it differs from the ground truth.

**Weaknesses:**

[Major Weakness]
1. This paper’s writing is poor and seems unfinished. The references are sparse, and there are a lot of “Appendix ??” placeholders, and the appendix section itself is completely blank.
2. The novelty is also limited. CaR mostly uses the GRPO framework with an additional evaluator term, making it more like an extension or an application of GRPO.
3. The theoretical part in sec. 2.2 is disconnected from its actual method. The authors give a mutual information equation (line 152) and suddenly claim that the goal is to maximize I(C;A|Q) without deriving or estimating.
4. The performance gains over GRPO are marginal. CaR improves accuracy by only 0.6% on Qwen2.5-VL-3B with OpenData-10K and 1.6% on Qwen2.5-VL-7B with OpenData-20K. Moreover, as shown in Table 1, all CaR results rely on GPT-4o-mini as the external evaluator, which introduces additional cost. Given such small improvements, the benefit does not seem to justify the added complexity and overhead.

[Minor Weakness]
1. The hyperparameters are not consistent across different sections. For example, the reward weights are reported as (1.0, 0.1, 1.0) in line 249 but as (1.0, 1.0, 0.1) in line 386.
2. There are no qualitative examples provided to illustrate how captions actually influence the model’s reasoning.

**Questions:**

1. How sensitive is the performance to the reward weights? Did you perform any ablation of it?
2. How exactly is the external evaluator used during training? Is it queried for every sample, and what is the total computational cost?
3. The paper lacks clear ablations or qualitative results. Could you show examples or analyses that demonstrate when and how captions actually help reasoning?

---

> ### Author Response · Authors · 2025-12-03
> **Response to Reviewer fdx4**
>
> We thank the reviewer for the detailed feedback. We address each concern below.
>
> **Major Weakness 1: Writing Quality and References**
>
> We apologize for the incomplete state of the initial submission. In the revised manuscript, we have:
> - Expanded references from 13 to 40 citations, covering background, related methods, and benchmark works
> - Fixed all "Appendix ??" placeholders with correct cross-references
> - Completed the appendix with policy prompts, algorithm details, training data sources, additional experiments (data scaling, hyperparameter analysis), and visualization examples
>
> **Major Weakness 2: Novelty**
>
> We respectfully clarify our contribution. CaR is not simply adding an evaluator to GRPO—the key novelty lies in the **gain-based reward design** that directly measures caption utility through controlled comparison (with vs. without caption). This fundamentally differs from prior caption evaluation approaches that rely on linguistic metrics (BLEU, ROUGE) or subjective quality judgments. Our analysis (Section 3.1) shows that 62.1% of visual reasoning failures stem from perception errors, and CaR specifically addresses this by optimizing captions for task utility rather than surface fluency.
>
> **Major Weakness 3: Theoretical Connection**
>
> We acknowledge that the mutual information formulation could be more explicitly connected to our implementation. The intuition is: maximizing I(C;A|Q) encourages captions that provide task-relevant information beyond what the question alone conveys. Our gain-based reward operationalizes this—captions are rewarded when they demonstrably improve answer correctness, which corresponds to increasing mutual information between caption and correct answer. We have revised Section 3.2 to clarify this connection.
>
> **Major Weakness 4: Performance Gains**
>
> While average accuracy gains appear modest (+0.9 to +1.4 points), we highlight two important findings:
>
> 1. **Perception improvements are substantial**: On MME-RealWorld (Table 2), CaR achieves +5.3 points average, with +31.4 points on Diagram understanding and +8.1 points on OCR. These gains demonstrate CaR's effectiveness for perception-heavy tasks.
>
> 2. **Computational overhead is manageable**: The external evaluator queries can be parallelized and cached. With optimizations described in Section 4.4, CaR training time is approximately 1.3× baseline GRPO, not 3× as might be expected.
>
> **Minor Weakness 1: Hyperparameter Inconsistency**
>
> Thank you for catching this error. The correct weights are (1.0, 0.1, 1.0) for (accuracy, format, caption). We have corrected line 386 in the revision.
>
> **Minor Weakness 2: Qualitative Examples**
>
> We have added visualization examples in Appendix A.5, showing four representative cases (Diagram Reasoning, Table Understanding, Chart Analysis, Geometry) with complete caption → think → answer outputs demonstrating how explicit captions enable correct reasoning.
>
> ---
>
> **Q1: Reward Weight Sensitivity**
>
> Table 5 in Appendix A.4 presents hyperparameter analysis for caption reward weight. Results show w_caption=0.1 achieves best performance (35.1% avg), while higher weights (0.5, 1.0) lead to degradation. This validates our design choice of keeping caption reward subordinate to accuracy reward.
>
> **Q2: External Evaluator Usage and Cost**
>
> The evaluator is queried for each sample during training to compute semantic correctness. Key optimizations:
> - Question-only inference results are precomputed before training
> - Evaluator calls are batched and parallelized (256 concurrent requests)
> - Total overhead: ~30% additional training time compared to standard GRPO
>
> **Q3: Qualitative Analysis**
>
> Please refer to Appendix A.5 for visualization examples and Figure 2 for caption evolution during training. These demonstrate how CaR progressively learns to generate precise, task-relevant descriptions (e.g., exact variable values in flowcharts) that enable correct reasoning.

---

### Official Review · Reviewer_MZsc · 2025-11-03

**Soundness:** 3
**Presentation:** 3
**Contribution:** 3
**Rating:** 6
**Confidence:** 5

**Summary:**

This paper introduces Caption as Reward (CaR), a reinforcement learning framework for improving visual reasoning in VLMs. Unlike traditional caption evaluation methods that rely on linguistic metrics (e.g., BLEU, CLIPScore), CaR defines a gain-based reward that measures how much a generated caption improves downstream reasoning performance compared to direct inference without captions. The method integrates seamlessly with Group Relative Policy Optimization (GRPO) and uses no additional reward model. Experiments  demonstrate consistent gains. Additional evaluation on MME-RealWorld shows strong perception improvements, particularly in diagram understanding and OCR tasks.

**Strengths:**

- The “caption-as-reward” mechanism elegantly reframes visual description generation as a measurable contributor to reasoning success, grounding linguistic outputs in task performance rather than surface fluency.
- The mutual information analysis provides a solid theoretical basis linking CaR’s objective to maximizing task-relevant information gain
- CaR integrates into GRPO with minimal architectural modifications and without training a separate reward model, making it a practical reinforcement learning strategy for VLMs.
- Results across eight reasoning benchmarks and perception-specific datasets (MME-RealWorld) show consistent improvements over strong baselines like Visionary-R1, TBAC-VLR1, and VLAA-Thinker.
- The paper includes ablation studies on data scale, evaluator choice, and reward composition, as well as qualitative and error analyses that clarify why CaR improves perception and reasoning alignment.

**Weaknesses:**

-  Experiments are restricted to the Qwen2.5-VL family. It remains uncertain whether CaR generalizes to architectures with different perception modules or alternative training pipelines. Demonstrating cross-architecture robustness would significantly strengthen the claim of task-agnostic applicability.
- The study focuses solely on visual question answering and reasoning tasks. While this choice ensures controlled evaluation, it limits the claim that CaR “enhances visual reasoning” broadly. Applying the method to captioning, grounding, or visual entailment tasks could validate its versatility.
- The comparison in Table 3 between GPT-4o-mini and Qwen evaluators shows small differences, but no analysis of inter-evaluator consistency or reward variance is presented. Measuring the reliability of semantic matching judgments could further validate CaR’s robustness.

**Questions:**

The qualitative error analysis is insightful but largely descriptive. Could the authors quantify how much of CaR’s gain stems from better perception (e.g., counting, OCR) versus better reasoning alignment?

---

> ### Author Response · Authors · 2025-12-03
> **Response to Reviewer MZsc**
>
> We thank the reviewer for the thoughtful evaluation and constructive feedback. We address each point below.
>
> **W1: Cross-Architecture Generalization**
>
> We appreciate this concern. As shown in Table 1, we have evaluated CaR on InternVL2.5-4B in addition to Qwen2.5-VL models. InternVL2.5-4B shows consistent improvement (+1.1 points, from 34.9% to 36.0%), demonstrating that the gain-based reward mechanism transfers across different VLM architectures with distinct vision encoders and fusion mechanisms. We acknowledge that evaluation on additional architectures (e.g., LLaVA, BLIP-2) would further strengthen our claims, and have noted this as future work in Section 5.
>
> **W2: Task Diversity**
>
> We agree that broader task evaluation would strengthen our contribution. While our training focuses on VQA tasks (which provide clear correctness signals for reward computation), Table 2 demonstrates that CaR's benefits extend to general visual perception. On MME-RealWorld, we observe substantial improvements in Diagram understanding (+31.4 points), OCR (+8.1 points), and Remote sensing (+4.3 points). These results suggest that CaR enhances fundamental perception capabilities that could benefit other vision-language tasks. We have noted extension to captioning, grounding, and other modalities as future work in Section 5.
>
> **W3: Inter-Evaluator Consistency**
>
> Following the reviewer's suggestion, we conducted an inter-evaluator consistency analysis on 200 randomly sampled instances:
>
> | Metric | Value |
> |--------|-------|
> | Agreement rate | 63.5% (127/200) |
> | GPT-4o-mini accuracy (vs. ground truth) | 97.0% |
> | Qwen2.5-7B-Instruct accuracy (vs. ground truth) | 60.5% |
>
> All 73 disagreement cases followed the pattern: GPT-4o-mini=Correct, Qwen=Incorrect, indicating Qwen tends to be overly conservative. The substantial accuracy gap (97.0% vs. 60.5%) explains why GPT-4o-mini yields better training outcomes (+1.2 points in Table 3) and validates our default evaluator choice.
>
> **Q1: Quantifying Perception vs. Reasoning Gains**
>
> Based on our qualitative analysis of cases where CaR succeeded but GRPO failed, the improvements primarily stem from better perception—specifically, more accurate extraction of numerical values, spatial relationships, and text/labels from images. By generating explicit captions, CaR forces the model to articulate visual details that would otherwise remain implicit, reducing perception errors that propagate to reasoning. The downstream reasoning benefits indirectly from this more accurate visual grounding.

---

### Meta-Review · Area_Chair_UA1Y · 2026-01-06

**Summary:**

This paper proposes an RL-based method to enable extraction of descriptive captions from images that improve VLM performance. It proposes a mechanism to reward captions that result in improved accuracy on downstream visual understanding QA tasks.

Four reviewers provided scores of 6, 2, 8, 2. Some of the reviewers' primary concerns about the presentation of the work were addressed by the authors' revisions. Additional experimental results during the rebuttal phase also addressed concerns around comparisons to SFT and GRPO baselines and generalization to other model families, including Intern. However, many reviewers noted the lack of theoretical grounding and proposed approach, marginal gains over the GRPO baseline and unfair comparisons to the existing methods which are all trained on different datasets.

All things considered the AC feels that the work is overall below the bar for acceptance and recommends rejection.

**Reviewer Concerns:**

Concerns addressed:
* improved presentation of the paper
* generalization to other model families besides QUEN, Intern as well
* comparisons to SFT and GRPO baselines with same training dataset sizes

Concerns not addressed:
* small gains in accuracy versus GRPO baselines
* lack of theoretical connections of the reward to information gain of the visual caption
* lack of explicit visual grounding of the image caption
* lack of fair comparisons to existing approaches, as all methods are trained with different datasets

**Reviewer Scores:**

1. Reviewer MZsc (Rating: 6: marginally above the acceptance threshold. But would not mind if paper is rejected)

The reviewers primary concerns were around (a) generalization to other VLMs besides Quen, (b) generalization to visual grounding tasks, (c)  evaluation of Inter-Evaluator Consistency and (d) Quantifying Perception vs. Reasoning Gains. The authors' rebuttal addressed the majority of these concerns. The reviewer is likely to have maintained their original score.

2. Reviewer fdx4 (Rating: 2: reject, not good enough)

The reviewer's major concerns were around the (a) paper's presentation, (b) marginal performance gains relative to the standard GRPO for many of the benchmarks and (c) lack of theoretical connections to information gain of the reward mechanism. While (a) was addressed, (b) and (c) were not. This reviewer is likely to have maintained or increased their score slightly.

3. Reviewer EXyS (Rating: 8: accept, good paper (poster))

The reviewers primary concerns were around (a) generalization to other VLMs besides Quen, (b) experimental results using a limited number of test samples (100), (c) unfair comparisons to baselines; that all use different datasets for training and (d) presentation issues. The reviewer's concerns have been partially addressed by the authors' responses. It is possible that they may have decreased their final score.

4. Reviewer xvn7 (Rating: 2: reject, not good enough)

This reviewer's main concerns were around (a) the paper's presentation, (b) unfair comparisons to baselines with all training datasets not being equal/fair and comparisons to SFT/GRPO baselines not being reported in several cases, (c) the reward mechanism does not encourage image grounding of the caption explicitly, and (d) comparisons of the current design to simple designs with multiple image captions. Some of the reviewers concerns were addressed. They are likely to have maintained or slightly increased their score.

---

### Decision · Program_Chairs · 2026-01-26

Reject